

# Irreducible representations of $\mathbb{Z}_2^2$-graded supersymmetry algebra and their applications

**Naruhiko Aizawa$^\star$**

Department of Physics, Osaka Metropolitan University, Sakai, Osaka 599-8531, Japan

$\star$ aizawa@omu.ac.jp

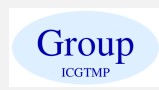

## Abstract

We give a brief review on recent developments of $\mathbb{Z}_2^n$-graded symmetry in physics in which hidden $\mathbb{Z}_2^n$-graded symmetries and $\mathbb{Z}_2^n$-graded extensions of known systems are discussed. This elucidates physical relevance of the $\mathbb{Z}_2^n$-graded algebras. As an example of physically interesting algebra, we take $\mathbb{Z}_2^2$-graded supersymmetry (SUSY) algebras and consider their irreducible representations (irreps). A list of irreps for $\mathcal{N} = 1, 2$ algebras is presented and as an application of the irreps, $\mathbb{Z}_2^2$-graded SUSY classical actions are constructed.

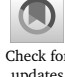

## 1 Introduction

It was more than half a century ago that Ree pointed out that one may generalize Lie algebras by grading with any abelian group [1]. The same object was rediscovered by Rittenberg and Wyler in late 70s [2, 3] (see also [4, 5]). Lie superalgebras are the simplest example of Ree's generalization where the abelian group is taken to be $\mathbb{Z}_2$. Among other possibilities, only $\mathbb{Z}_2^n := \mathbb{Z}_2 \times \cdots \times \mathbb{Z}_2$ ($n$ times) allows us to determine the generalized Lie algebra in terms of commutators and anticommutators so that the $\mathbb{Z}_2^n$-graded algebra is a natural generalization of Lie superalgebras [2, 3]. This implies that the $\mathbb{Z}_2^n$-graded algebras have potential applicability to physical problems, but it is hard to say that such algebra itself is widely recognized in physics community.

One of the purposes of this paper is to emphasize, by providing a brief review on recent developments of $\mathbb{Z}_2^n$-graded symmetry in physical problems, that the $\mathbb{Z}_2^n$-graded algebras are *not* unusual in physics. Rather, they are ubiquitous and would be an important notion to understand nature. This is the contents of §2. The second purpose is to present irreducible representations of $\mathbb{Z}_2^2$-graded version of the supersymmetry algebra [6, 16] which is an algebra of recent particular interest. Here the SUSY-algebra means the super-Poincaré algebra in $(0 + 1)$D spacetime. Knowledge on irreps of an algebra is of fundamental importance for its

physical and mathematical applications. In §3.1, irreps of $\mathbb{Z}_2^2$-graded SUSY algebra of $\mathcal{N} = 1, 2$ are presented. As an application of the irreps, we consider $\mathbb{Z}_2^2$-graded classical mechanics in §3.2. A $\mathbb{Z}_2^2$-graded SUSY transformation is defined by the irrep and classical actions invariant under the transformation and conserved Noether charges are given explicitly.

# 2 $\mathbb{Z}_2^n$-graded algebras in physics

This section is a very brief review on recently observed relations between $\mathbb{Z}_2^n$-graded algebras and physics. For those discussed in earlier days, readers may refer the references in [7].

## 2.1 $\mathbb{Z}_2^n$-graded algebras in known systems

(a) symmetries of Lévy-Leblond equation [8, 9].

The Lévy-Leblond equation is a quantum mechanical wave equation describing a spin $1/2$ particle in non-relativistic setting. The wave function is a four-component spinor which reproduces, when coupled with electromagnetic field, gyromagnetic ratio two as the Dirac equation does. The equation has Galilean superconformal symmetry, but there exit other symmetry generators which never close in a superalgebra but in $\mathbb{Z}_2^2$-graded algebra. Thus, the symmetry of the equation is given by a $\mathbb{Z}_2^2$-graded algebra.

Similar situation is also observed in the supersymmetric harmonic oscillator discussed in [10]. The system has additonal symmetry generators to the ones in [10] and the whole symmetry generators close in a $\mathbb{Z}_2^2$-graded algebra. This is one of the examples that even simple systems have $\mathbb{Z}_2^2$-graded symmetries.

(b) mixed system of parabosons and parafermions [11].

It is known that parafermion and paraboson algebras are isomorphic to orthogonal algebra and orthosymplectic superalgebra, respectively. There are two possible ways of mixing parabosons and parafermions and form an larger algebra. It has been known that one of them is isomorphic to the superalgebra $osp(2m + 1|2n)$. Recently, it was shown that the another one is isomorphic to a $\mathbb{Z}_2^2$-graded extension of orthosymplectic superalgebra. Using this fact, a Fock representation of $\mathbb{Z}_2^2$-graded orthosymplectic superalgebras has been constructed in [12].

(c) Clifford algebras [3, 13, 14].

Clifford algebra $Cl(n, m)$ is regarded as a $\mathbb{Z}_2^n$-graded algebra. This identification is not unique, namely, there are several different ways of assigning $\mathbb{Z}_2^n$-grading to the Clifford algebras. Quaternion and split-quaternion realize $Cl(0, 3)$ and $Cl(2, 1)$, respectively. Thus, they are also $\mathbb{Z}_2^n$-graded algebra. One may use this fact to realize a $\mathbb{Z}_2^n$-graded algebra in terms of an ordinary superalgebra and a Clifford algebra which leads us to $\mathbb{Z}_2^n$-graded extensions of supersymmetric and superconformal quantum mechanics (see §2.2).

These observations reveal hidden $\mathbb{Z}_2^n$-graded algebraic structure in well-known systems. What is remarkable is that $\mathbb{Z}_2^n$-graded algebras are found in simpler systems compared with the earlier works where SUGRA, string theory were discussed. We believe that this is an illustration of the fact that the $\mathbb{Z}_2^n$-graded algebras are found in many places in physics so that they would play certain roles for deeper understanding of nature.

## 2.2 $\mathbb{Z}_2^n$-graded extensions of known systems

(a) supersymmetric and superconformal quantum mechanics [15–18].

Supersymmetric quantum mechanics (SQM) can be generalized to $\mathbb{Z}_2^n$-graded setting for arbitrary values of $n$. It is also possible to have $\mathbb{Z}_2^n$-graded extensions of many models of superconformal mechanics (SCM). This is a consequence of $\mathbb{Z}_2^n$-graded algebraic nature of the

Clifford algebras. One may find an appropriate combination of SQM or SCM and a Clifford algebra which produce its $\mathbb{Z}_2^2$-graded extension. What is remarkable is that these extensions are not unique. That is, for a given model of SQM or SCM, we may have several inequivalent $\mathbb{Z}_2^n$-graded extensions.

(b) supersymmetric classical systems [19–23].

Several $\mathbb{Z}_2^2$-supersymmetric classical actions (field theory, mechanics), which produce $\mathbb{Z}_2^2$-supersymmetric quantum systems upon quantization, have been proposed. Contrast to the quantum mechanics, only $\mathbb{Z}_2^2$-grading is considered in the literature. The actions are constructed by extending $D$-module presentation and superfield formulation to $\mathbb{Z}_2^2$-setting. Extension of superfield formalism is not straightforward since $\mathbb{Z}_2^2$-graded superspace has an extra bosonic coordinate which is not nilpotent and anticommute with fermionic coordinate so that the superfield is a formal power series in this exotic bosonic coordinate. Furthermore, integration on the $\mathbb{Z}_2^n$-graded superspace is highly non-trivial and only $\mathbb{Z}_2^2$ case is known yet [24].

(c) sine-Gordon equation [25].

It has been shown that a $\mathbb{Z}_2^2$-graded extension of the sine-Gordon equation is solvable. This suggest the existence of a new class of integrable systems which are characterized by $\mathbb{Z}_2^2$-graded symmetry.

(d) detectability of $\mathbb{Z}_2^2$-graded supersymmetry [26, 27].

It is an important question whether the $\mathbb{Z}_2^2$-graded supersymmetry is *physically* different from the ordinary one. The question has been answered affirmatively. Existence of operators which distinguish $\mathbb{Z}_2^2$ and $\mathbb{Z}_2$-graded SQM was shown in multipartite sector of a simple model with harmonic oscillator potential. $\mathbb{Z}_2^2$-graded SUSY describes a kind of para-particle since, by definition of the symmetry algebra, it has commuting fermions and exotic bosons. Recently, it was reported that para-particle oscillators were realized experimentally [28]. This implies the possibility of experimental realization of $\mathbb{Z}_2^2$-graded para-particles.

(e) super division algebras [29].

It is known that the number of inequivalent associative real super division algebras is ten. Three of them are purely even ($\mathbb{R}, \mathbb{C}$ and $\mathbb{H}$) and another seven with odd elements. These ten super division algebras have a deep connection with other objects having ten inequivalent classes such as the periodic table of topological insulators and superconductors, Morita equivalence classes of real and complex Clifford algebras, and classical families of compact symmetric spaces (see [30] for a review).

The superdivision algebra is a $\mathbb{Z}_2$-graded extension of the purely even one. There is no reason to stop at $\mathbb{Z}_2$-grading, one may consider $\mathbb{Z}_2^n$-graded division algebras and would expect surprising connection with other objects. In [29], a classification of $\mathbb{Z}_2^2$-graded division algebras has been done. It was shown that there are 13 $\mathbb{Z}_2^2$-graded division algebras in addition to the $\mathbb{Z}_2$-graded counterpart. Objects having connection with these $\mathbb{Z}_2^2$-graded division algebras are not known yet. Finding them is an exciting problem.

From these observation, one may conclude that $\mathbb{Z}_2^n$-graded symmetries enlarge the concepts of physical importance and open up new fields of research interest.

# 3 Irreducible representations of $\mathbb{Z}_2^2$-graded SUSY algebras

## 3.1 Irreps of $\mathcal{N} = 1, 2$ algebras

In this section, we present irreps of $\mathbb{Z}_2^2$-graded SUSY algebras. We deal with $\mathcal{N} = 1$ and $\mathcal{N} = 2$ algebras and consider their representations in a $\mathbb{Z}_2^2$-graded vector space.

Let us first recall the definition of $\mathbb{Z}_2^2$-graded algebra [1–3]. It is a direct sum of four vector spaces each of which is labeled by an element of $\mathbb{Z}_2^2 : \mathfrak{g} = \mathfrak{g}_{(0,0)} \oplus \mathfrak{g}_{(1,0)} \oplus \mathfrak{g}_{(0,1)} \oplus \mathfrak{g}_{(1,1)}$. The multiplication of two elements $X_{\vec{a}} \in \mathfrak{g}_{\vec{a}}$ and $Y_{\vec{b}} \in \mathfrak{g}_{\vec{b}}$ is defined by generalized Lie bracket $[\![\ ,\ ]\!] : \mathfrak{g} \times \mathfrak{g} \to \mathfrak{g}$ which is a bilinear map and satisfies

$$[\![ X_{\vec{a}}, Y_{\vec{b}} ]\!] = -(-1)^{\vec{a} \cdot \vec{b}} [\![ Y_{\vec{b}}, X_{\vec{a}} ]\!] \in \mathfrak{g}_{\vec{a}+\vec{b}}, \tag{1}$$

$$[\![ X_{\vec{a}}, [\![ Y_{\vec{b}}, Z_{\vec{c}} ]\!] ]\!] = [\![ [\![ X_{\vec{a}}, Y_{\vec{b}} ]\!], Z_{\vec{c}} ]\!] + (-1)^{\vec{a} \cdot \vec{b}} [\![ Y_{\vec{b}}, [\![ X_{\vec{a}}, Z_{\vec{c}} ]\!] ]\!], \tag{2}$$

where $\vec{a} \cdot \vec{b}$ is the inner product of two-component vectors. The relation (1) implies that the generalized Lie bracket is realized by commutator and anticommutator. The relation (2) is a $\mathbb{Z}_2^2$-graded version of the Jacobi identity.

Now we turn to the $\mathbb{Z}_2^2$-graded SUSY algebra [6, 16]. The $\mathcal{N} = 1$ algebra is of four dimension

$$H \in \mathfrak{g}_{(0,0)}, \qquad Q_{10} \in \mathfrak{g}_{(1,0)}, \qquad Q_{01} \in \mathfrak{g}_{(0,1)}, \qquad Z \in \mathfrak{g}_{(1,1)}, \tag{3}$$

and the relations is given, in terms of commutator and anticommutator, by

$$\begin{aligned} \{Q_{01}, Q_{01}\} = \{Q_{10}, Q_{10}\} = 2H, \qquad [Q_{01}, Q_{10}] = 2iZ, \\ [H, Q_{01}] = [H, Q_{10}] = 0, \qquad [Z, H] = \{Z, Q_{01}\} = \{Z, Q_{10}\} = 0. \end{aligned} \tag{4}$$

While, the $\mathcal{N} = 2$ algebra is six-dimensional

$$H \in \mathfrak{g}_{(0,0)}, \qquad Q_{10}, Q_{10}^\dagger \in \mathfrak{g}_{(1,0)}, \qquad Q_{01}, Q_{01}^\dagger \in \mathfrak{g}_{(0,1)}, \qquad Z \in \mathfrak{g}_{(1,1)}. \tag{5}$$

The defining relations are

$$\begin{aligned} \{Q_a, Q_a^\dagger\} = H, \qquad [Q_{01}, Q_{10}^\dagger] = [Q_{01}^\dagger, Q_{10}] = iZ, \\ \{Q_a, Q_a\} = \{Q_a^\dagger, Q_a^\dagger\} = \{Z, Q_a\} = \{Z, Q_a^\dagger\} = [Z, H] = 0, \\ [Q_a, Q_b] = [Q_a^\dagger, Q_b^\dagger] = 0, \quad a, b = (1, 0), (0, 1). \end{aligned} \tag{6}$$

One may see that $H$ is central, while $Z$ is $\mathbb{Z}_2^2$-graded centeral. It follows that $H, Z$ span the Cartan subalgebra, but they are not diagonalizable simultaneously as they belong to the subspaces of different $\mathbb{Z}_2^2$-degree. It is easy to see that $H^2$ and $Z^2$ are the second order Casimir which commute with all the elements. Thus, irreps are labeled by their eigenvalues which are denoted by $E^2$ for $H^2$ and $\lambda$ for $Z^2$. The variables $E, \lambda$ also labels irreps of the Cartan subalgebra in a $\mathbb{Z}_2^2$-graded representation space [7]:

**Lemma 1.** *Irreps of the Cartan subalgebra spanned by $H, Z$ are equivalent to one of the followings:*

1. *One dimensional irrep $v(E, 0) = $ lin. span $\langle v_0 \rangle$*

$$H v_0 = E v_0, \qquad Z v_0 = 0. \tag{7}$$

2. *Two dimensional irrep $v(E, \lambda) = $ lin. span $\langle v_0, v_1 \rangle$, $\lambda \neq 0$*

$$H v_0 = E v_0, \qquad v_1 = Z v_0, \qquad Z v_1 = \lambda v_0. \tag{8}$$

*Without loss of generality, one may assume the $v_0$ belongs to the subspace of $\mathbb{Z}_2^2$-degree $(0, 0)$.*

Representations of the $\mathbb{Z}_2^2$-graded SUSY algebras are induced from $v(E, \lambda)$, but the induced representations are *not* irreducible. For $\mathcal{N} = 1, 2$ cases, due to their simplicity, one may find

invariant subspaces explicitly in the induced representation space. For instance, the induced space for $\mathcal{N} = 1, \lambda \neq 0$ is eight dimensional whose basis is taken to be

$$
\begin{array}{cccc}
v_0, & Q_{10}v_0, & Q_{01}v_0, & \frac{1}{2}\{Q_{10},Q_{01}\}v_0, \\
v_1, & Q_{10}v_1, & Q_{01}v_1, & \frac{1}{2}\{Q_{10},Q_{01}\}v_1.
\end{array}
\tag{9}
$$

The four dimensional invariant subspace is spanned by the vectors

$$
v_{00} := \alpha v_0 + \frac{\beta}{2}\{Q_{10},Q_{01}\}v_1, \qquad v_{11} = Zv_{00}, \qquad v_{10} = Q_{10}v_{00}, \qquad v_{01} = Q_{01}v_{00}, \tag{10}
$$

provided that $\alpha, \beta$ satisfy the relations

$$
\alpha^2 = \lambda\beta^2(E^2 - \lambda), \qquad (Ec + i\lambda)^2 = \lambda(E^2 - \lambda), \tag{11}
$$

where $c$ is the constant connecting two vectors : $Q_{01}Zv_{00} = cQ_{10}v_{00}$. In the case of $\mathcal{N} = 2$ algebra, the induced representation is 16 and 32 dimensional for $\lambda = 0$ and $\lambda \neq 0$, respectively. Like the $\mathcal{N} = 1$ case, one may find a basis of irreps explicitly. Details are presented in [22,23]. We come to present a list of irreps of the $\mathbb{Z}_2^2$-graded SUSY algebras:

**Theorem 2.** *The $\mathcal{N} = 1$ algebra has a 4D irrep for all possible values of $E, \lambda$. While, the $\mathcal{N} = 2$ algebra has some inequivalent irreps depending the value of $E, \lambda$ :*

1. *four inequivalent 4D irreps if $\lambda = 0$.*

2. *two inequivalent 4D irreps if $\lambda = E^2$.*

3. *two inequivalent 8D irreps if $\lambda \neq 0$ and $\lambda \neq E^2$.*

Some remarks are in order. If $\lambda = 0$, then $Z$ is represented by the zero matrix so that the algebra is almost two copies of the ordinary SUSY algebra. "Almost" means that two supercharges $Q_{10}$ and $Q_{01}$ commute instead of anticommute which is the case of ordinary SUSY. If $\lambda = E^2$, irrep of both $\mathcal{N} = 1$ and $\mathcal{N} = 2$ algebras is four dimensional and this irrep is peculiar since $H^2 = Z^2$ holds for this irrep. In fact, all the physical models discussed in the literature (see §2.2) is limited to this particular irrep. Our theorem shows the existence of wider irreps, the restriction $\lambda = E^2$ is not necessary for both $\mathcal{N} = 1$ and $\mathcal{N} = 2$ algebras. Therefore, we would expect the existence of physical models in which irreps with $\lambda \neq E^2$ are realized.

## 3.2 $\mathbb{Z}_2^2$-graded SUSY classical mechanics

As an application of the irreps of §3.1, we construct $\mathbb{Z}_2^2$-graded SUSY actions of classical mechanics. We employ the four dimensional irrep ($\lambda = E^2$) of $\mathcal{N} = 2$ algebra. The representation basis is taken to be four complex variables $x(t), z(t), \psi(t), \xi(t)$ which are functions of time $t$. Their $\mathbb{Z}_2^2$ degree are $(0,0), (1,1), (1,0), (0,1)$, respectively and we assume that they are $\mathbb{Z}_2^2$-commutative: $[\![A,B]\!] = 0$. This assignment makes $x(t)$ an ordinary complex number, $\psi(t), \xi(t)$ nilpotent and $z(t)$ anticommute with $\psi(t), \xi(t)$. Thus, $x(t)$ is a bosonic variables, $\psi(t), \xi(t)$ are fermionic and $z(t)$ is an exotic bosonic.

The action of the $\mathcal{N} = 2$ algebra on this basis defines a $\mathbb{Z}_2^2$-graded SUSY transformation which reads as follows:

$$
\begin{array}{ll}
Q_{10} : (x,z,\psi,\xi) \to (\psi,\xi,i\dot{x},i\dot{z}), & (\bar{x},\bar{z},\bar{\psi},\bar{\xi}) \to (-\bar{\psi},\bar{\xi},-i\dot{\bar{x}},i\dot{\bar{z}}), \\
Q_{01} : (x,z,\psi,\xi) \to (-i\xi,-i\psi,-\dot{z},-\dot{x}), & (\bar{x},\bar{z},\bar{\psi},\bar{\xi}) \to (-i\bar{\xi},i\bar{\psi},\dot{\bar{z}},-\dot{\bar{x}}), \\
Z : (x,z,\psi,\xi) \to (-\dot{z},-\dot{x},\dot{\xi},\dot{\psi}), & (\bar{x},\bar{z},\bar{\psi},\bar{\xi}) \to (-\dot{\bar{z}},-\dot{\bar{x}},-\dot{\bar{\xi}},-\dot{\bar{\psi}}),
\end{array}
\tag{12}
$$

where the bar indicates the complex conjugation and $H$ transform all variables to their time derivative.

One may easily write down an action invariant under the transformation (12):

$$L_0 = \dot{\bar{x}}\dot{x} + \dot{\bar{z}}\dot{z} - i(\bar{\psi}\dot{\psi} + \bar{\xi}\dot{\xi}). \tag{13}$$

This is a free theory with four complex dynamical variables and interaction will be introduced by the way similar to [20]. It is also easy to compute the Noether charges. With the notation same as the symmetry generators, they are given by

$$
\begin{aligned}
H &= \dot{\bar{x}}\dot{x} + \dot{\bar{z}}\dot{z}, \quad Q_{10} = \dot{x}\bar{\psi} + \dot{z}\bar{\xi}, \quad Q_{10}^\dagger = \dot{\bar{x}}\psi - \dot{\bar{z}}\xi, \\
Z &= \dot{\bar{x}}\dot{z} + \dot{x}\dot{\bar{z}}, \quad Q_{01} = \dot{x}\bar{\xi} + \dot{z}\bar{\psi}, \quad Q_{01}^\dagger = \dot{\bar{x}}\xi - \dot{\bar{z}}\psi.
\end{aligned}
\tag{14}
$$

Like the standard supersymmetry, it is possible to convert dynamical variables to auxiliary ones. We give three examples of such conversion together with the Noether charges.

1) Define $F := \dot{z}$, $\overline{F} := \dot{\bar{z}}$, then all the degree $(1,1)$ variables become auxiliary:

$$L_0 \ \rightarrow \ L_1 = \dot{\bar{x}}\dot{x} + |F|^2 - i(\bar{\psi}\dot{\psi} + \bar{\xi}\dot{\xi}), \tag{15}$$

$$H = \dot{\bar{x}}\dot{x}, \quad Z = 0, \quad Q_{10} = \dot{x}\bar{\psi}, \quad Q_{10}^\dagger = \dot{\bar{x}}\psi, \quad Q_{01} = \dot{x}\bar{\xi}, \quad Q_{01}^\dagger = \dot{\bar{x}}\xi. \tag{16}$$

2) Define $y := \frac{1}{2}(x + \bar{x})$, $A := \frac{i}{2}(\dot{x} - \dot{\bar{x}})$, then one of the degree $(0,0)$ variables becomes auxiliary:

$$L_0 \ \rightarrow \ L_2 = \dot{y}^2 + A^2 + \dot{\bar{z}}\dot{z} - i(\bar{\psi}\dot{\psi} + \bar{\xi}\dot{\xi}), \tag{17}$$

$$
\begin{aligned}
H &= \dot{y}^2 + \dot{\bar{z}}\dot{z}, \quad Q_{10} = \dot{y}\bar{\psi} + \dot{z}\bar{\xi}, \quad Q_{10}^\dagger = \dot{y}\psi - \dot{\bar{z}}\xi, \\
Z &= \dot{y}(\dot{z} + \dot{\bar{z}}), \quad Q_{01} = \dot{y}\bar{\xi} + \dot{z}\bar{\psi}, \quad Q_{01}^\dagger = \dot{y}\xi - \dot{\bar{z}}\psi.
\end{aligned}
\tag{18}
$$

3) Define $a = \dot{x}$, $\bar{a} = \dot{\bar{x}}$, then all the degree $(0,0)$ variables become auxiliary:

$$L_0 \ \rightarrow \ L_3 = |a|^2 + \dot{\bar{z}}\dot{z} - i\left(\bar{\psi}\dot{\psi} + \bar{\xi}\dot{\xi}\right), \tag{19}$$

$$H = \dot{\bar{z}}\dot{z}, \quad Z = 0, \quad Q_{10} = \dot{z}\bar{\xi}, \quad Q_{10}^\dagger = \dot{\bar{z}}\xi, \quad Q_{01} = \dot{z}\bar{\psi}, \quad Q_{01}^\dagger = \dot{\bar{z}}\psi. \tag{20}$$

An interesting observation is the degree $(1,1)$ charge $Z$ vanishes if all degree $(0,0)$ variables or all degree $(1,1)$ ones are auxiliary. This is understood from the form of $Z$ given in (14). More details of classical actions for $\mathcal{N} = 1, 2$ are found in [22, 23].

# 4 Conclusion

Based on many observations of $\mathbb{Z}_2^2$-graded symmetry in physics, we asserted physical relevance of $\mathbb{Z}_2^2$-graded algebras. Supersymmetry is one of the most important symmetries in physics so that its $\mathbb{Z}_2^2$-extensions were considered. We present irreps of $\mathbb{Z}_2^2$-graded SUSY algebras of $\mathcal{N} = 1, 2$ on $\mathbb{Z}_2^2$-graded representation space. Even though our investigation was restricted to (0+1)D spacetime, due to the $\mathbb{Z}_2^2$-grading of the representation space, the algebras have richer irreps compared with the standard SUSY algebra. We use the four dimensional irrep of $\mathcal{N} = 2$ algebra to construct $\mathbb{Z}_2^2$-graded classical mechanics and discuss its property such as vanishing Noether charge of $\mathbb{Z}_2^2$-degree $(1,1)$.

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
