# Peer review of "Irreducible representations of $\mathbb{Z}_2^2$-graded supersymmetry algebra and their applications"

_SciPost Physics Proceedings, doi:SciPost Phys. Proc. 14, 016 (2023)_

## Round 1 · Referee Report · Anonymous (Referee 1) · 2022-12-23

Strengths

  1. The paper contains a very nice review of applications of $Z_2^2$ and more generaly $Z_2^n$ graded algebra in physics. In particular the connection with symmetries of Levy-Leblond equation, mixed parabosons and parafermions and clifford algebras. The section 2.2 contain further context in which they have applications.

  2. The subsection 3.1 present details of the irreducible representations. The results seems correct and are nicely summarized.

  3. The subsection 3.2 is particularly relevant for this journal as it provide some insight into construction of Z_2^2 graded SUSY classical mechanics.

  4. The paper is well written and has appropriate references.

Weaknesses

I don't see obvious weaknesses in this paper. The author has contributed to this area quite actively in recent years and the paper is a nice addition to this proceeding which summarises results in the area.

Report

I recommend the paper to be accepted for publication in this journal as it meet the criteria.

---

## Editorial Decision

published